# `NeurVLA`: Unleashing Failure-Handling Capability of Vision-Language-Action Models via Neural-Symbolic Reasoning

Xuqi Liu [1] [*]  Minghe Gao [1] [*]  Juncheng Li [1] [✉]  Siliang Tang [1]

## Abstract

Vision-Language-Action models have recently shown promising progress in embodied robotic manipulation, yet their generalization to diverse open-ended embodied tasks is often hindered by execution failures. While prior work has explored failure handling, existing approaches still suffer from two fundamental limitations: coarse-grained failure correction and unreliable failure prevention. These limitations lead to brittle decision-making when VLA models are deployed in novel tasks and environments. To address them, we propose `NeurVLA`, a neural-symbolic framework that jointly addresses failure correction and prevention via neural-symbolic reasoning and further internalizes these failure-handling capabilities into VLA models. Experiments demonstrate that `NeurVLA` achieves strong performance and robust generalization across diverse tasks.

## 1. Introduction

In recent years, Vision-Language-Action (VLA) models have made significant progress in embodied robotic manipulation tasks (Kim et al., 2024; Zheng et al., 2024; Qu et al., 2025). By jointly modeling visual observations and language instructions, VLA models can directly generate executable robot actions to perform tasks end to end. However, the practical utility of VLA models depends not merely on achieving high success rates on seen tasks, but on their ability to generalize to diverse tasks and continually changing environments. Due to the open-ended nature of embodied tasks, characterized by variability in task configurations, object properties, and environmental structures, VLA models inevitably encounter execution failures when deployed in unseen tasks or environments (Gu et al., 2025; Lin et al., 2025; Yu et al., 2025). To operate effectively under such

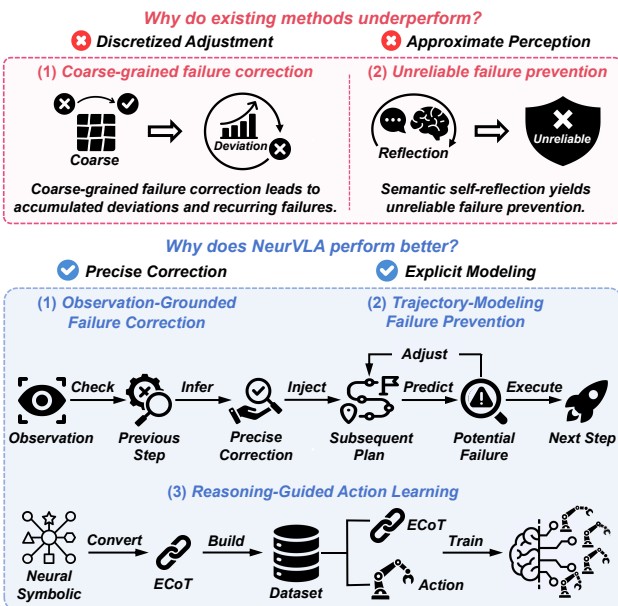

Figure 1. `NeurVLA` is a framework that overcomes core limitations of existing VLA methods, namely coarse-grained failure correction and unreliable failure prevention, and further internalizes failure-handling capability into VLA model.

uncertainty, VLA models must adapt their actions through iterative trial and failure (Zang et al., 2025; Bai et al., 2025; Li et al., 2025). Consequently, the capability of VLA models to correct failures and prevent potential failures is essential for robust generalization and reliable deployment.

While recent studies have begun to enhance the failure-handling capability of VLA models, existing approaches still suffer from two closely interconnected issues: **(1) Coarse-grained failure correction.** Approaches like FPC-VLA (Yang et al., 2025b) perform failure correction by invoking a visual-language model (VLM) to produce discretized direction and magnitude adjustments. However, the absence of precise localization and geometric analysis prevents the model from generating fine-grained corrective actions grounded in the actual spatial configuration. As a result, such corrections often introduce residual deviations that accumulate over subsequent steps and ultimately lead to repeated failures or task breakdown. **(2) Unreliable failure**

[*]Equal contribution [1]Zhejiang University, Hangzhou, China. Correspondence to: Juncheng Li <junchengli@zju.edu.cn>.

*Proceedings of the 43rd International Conference on Machine Learning*, Seoul, South Korea. PMLR 306, 2026. Copyright 2026 by the author(s).

**prevention.** Methods such as CollabVLA (Sun et al., 2025) rely primarily on semantic self-reflection from a VLM to identify uncertainty and potential failures. However, the lack of explicit future motion-trajectory modeling and geometric hazard checking limits the calibration of failure prevention, particularly in complex scenes, potentially leading to missed hazards or unnecessary interventions. These limitations are summarized in Figure 1.

A key observation underlying these limitations is that fine-grained failure correction requires precise analysis of object states from observations, while reliable failure prevention demands explicit modeling and verification of action-induced state transitions. However, current VLA methods lack such capabilities, and therefore rely on approximate and weakly grounded VLM-based judgments to guide failure handling, which undermines generalization to diverse tasks and environments. Inspired by this observation, we propose **NeurVLA**, which leverages a programmatic paradigm to invoke specialized models for precise observation analysis and explicit trajectory prediction of action-induced state transitions. Based on the processes, **NeurVLA** derives introspective Embodied Chain-of-Thoughts (ECoTs) and uses them to train VLA models, thereby internalizing fine-grained failure correction and reliable failure prevention as intrinsic model capabilities. Specifically, **NeurVLA** combines sequential and complementary components in a unified framework. **(1) For failure correction,** **NeurVLA** employs *Observation-Grounded Failure Correction*, generating executable code to precisely analyze object poses, verify the completion of previous steps, and inject fine-grained corrective steps into the subsequent plan when failures are detected. **(2) For failure prevention,** conditioned on the subsequent plan, **NeurVLA** adopts *Trajectory-Modeling Failure Prevention*, programmatically and explicitly modeling action-induced motion trajectories, identifying potential failures, and iteratively refining the plan until no failure is predicted. Subsequently, **NeurVLA** executes the plan using policy code to interact with the environment.

Moreover, to learn and internalize the above failure-handling capabilities within the VLA model rather than relying on external intervention such as FailSafe (Lin et al., 2025), **NeurVLA** further introduces *Reasoning-Guided Action Learning*, which converts the above reasoning processes into introspective ECoTs and leverages supervised fine-tuning (SFT) and contrastive learning to support introspective reasoning under different failure conditions. By combining failure-handling capability construction and ECoT-based capability learning, **NeurVLA** enables VLA models to acquire fine-grained failure correction and reliable failure prevention capabilities, supporting generalization across diverse scenarios.

Notably, similar to prior programmatic approaches, **NeurVLA** can adapt to a broad range of embodied tasks beyond manipulation, since a robot's behavior can be controlled through calls to control primitive APIs, and embodied spatial reasoning can be represented as an executable program composed of modular functions. Moreover, the scalability of **NeurVLA** stems from the modularity of its control, perception, and prediction components—by simply updating or adding API extensions, it can seamlessly integrate diverse embodiments and extend to new domains without altering the overall architecture.

Extensive experiments demonstrate that **NeurVLA** can be seamlessly integrated with various VLA architectures. It consistently achieves strong performance across multiple simulated benchmarks, improving average success rates by 12.2 points over the backbone, and also delivers significant gains in real-world experiments with a 51.9 points increase. Moreover, it exhibits strong robustness under different modality perturbations, improving task success rates by 12.3 points compared to other baselines and demonstrating higher tolerance to increasing noise levels. It introduces modest computational overhead, making it capable of robust generalization and deployment in real-world scenarios.

**In summary, our contributions are as follows:**

- We propose **NeurVLA**, a reasoning framework that jointly addresses failure correction and prevention for VLA models by integrating *Observation-Grounded Failure Correction* with *Trajectory-Modeling Failure Prevention*.

- **NeurVLA** further proposes *Reasoning-Guided Action Learning* to internalize the failure-handling capability derived from the above reasoning processes in VLA models.

- Extensive experiments demonstrate that **NeurVLA** consistently achieves strong performance and robust generalization across diverse embodied tasks.

## 2. Related Work

### 2.1. Failure Handling on Robotic Manipulation

To address execution failure challenges, recent works have explored failure detection, correction, and prevention. For example, AHA (Duan et al., 2024) and REFLECT (Liu et al., 2023b) focus on identifying and explaining failures but do not provide executable corrective actions. FPC-VLA introduces failure correction but relies on coarse-grained corrective actions. In addition, many approaches, including RoboReflect (Luo et al., 2025), RACER (Dai et al., 2025), and RoboFAC (Lu et al., 2025), lack mechanisms for preventing potential failures. CollabVLA further attempts failure prevention through qualitative VLM-based

judgments, resulting in unreliable prevention in complex scenarios. Finally, approaches such as FailSafe and YAY Robot (Shi et al., 2024) handle failures through auxiliary recovery models or human intervention, without internalizing failure-handling capability as an intrinsic property of the VLA model. In contrast, **NeurVLA** jointly supports fine-grained failure correction, reliable failure prevention, and the internalization of failure-handling capability.

## 2.2. Embodied CoT for VLA Models

Recent works (Zawalski et al., 2024; Zhao et al., 2025; Huang et al., 2025b; Yuan et al., 2025) have explored ECoT to improve the performance of VLA models. By generating intermediate thoughts before actions, ECoT can handle complex tasks. Moreover, HyT (Mazzaglia et al., 2025) enables VLA models to learn from ECoT during training while optionally omitting it at inference time, transferring the benefits of ECoT without incurring time overhead.

## 3. Method

**NeurVLA** constructs and internalizes robust failure-handling capabilities in a collaborative manner. As shown in Figure 2, **NeurVLA** divides the task completion process into several iterations, where each iteration consists of generating a reasoning process and executing a semantically meaningful action. In each iteration, given the current observation $o$ and the task instruction $inst$, **NeurVLA** first assesses whether the previous action has been successfully completed based on the observation, and injects fine-grained corrective steps into the subsequent plan when necessary (Sec. 3.1). It then analyzes the updated plan by explicitly modeling action-induced motion trajectories and potential collisions, iteratively refining the plan until safety is verified (Sec. 3.2). After verification, **NeurVLA** translates the first step of the subsequent plan into a policy code block to interact with the environment. Subsequently, **NeurVLA** converts the above reasoning processes into introspective ECoTs, aligns them with the executed actions to yield ECoT-action instances, and performs SFT and contrastive learning on these instances to internalize failure correction and prevention capabilities (Sec. 3.3).

### 3.1. Observation-Grounded Failure Correction

In embodied tasks, execution failures are inevitable due to imprecise perception, control noise, and unexpected environmental changes. Existing failure correction approaches typically rely on coarse-grained adjustments, which often result in insufficient correction, accumulated deviations, or repeated failures. To address this challenge, **NeurVLA** introduces *Observation-Grounded Failure Correction* ($C_{OG}$) to precisely analyze post-execution observations and inject fine-grained corrective steps into subsequent plans when

failures are detected.

Specifically, **NeurVLA** constructs an enhanced version of visual programming (VP) (Gupta & Kembhavi, 2023) by augmenting standard VP with geometry functions for spatial relation reasoning and metric distance computation, defined in the same coordinate system as Omni3D (Brazil et al., 2023). In each iteration, **NeurVLA** instantiates an executable program $p_c$ based on the enhanced VP, and uses it to analyze the current observation to assess whether the previous action has been successfully completed. During execution, $p_c$ produces a collection of intermediate variables, which are organized as a dictionary and denoted as $d_c$. When planar reasoning suffices, this assessment is carried out by executing standard VP over the observation; when explicit spatial reasoning is required, **NeurVLA** invokes 3D-MOOD (Yang et al., 2025c), an open-set monocular 3D object detection model, to infer the 3D coordinates of task-relevant objects and the gripper, and evaluates their spatial relationships through geometric computation. Benefiting from executable and coordinate-grounded analysis, **NeurVLA** can faithfully verify action completion and produce precise, metric-level corrective steps that are directly conditioned on the actual execution state. These fine-grained corrective steps are injected into the subsequent plan to guide continued task execution.

### 3.2. Trajectory-Modeling Failure Prevention

Based on the updated subsequent plan produced by failure correction, **NeurVLA** aims to reduce execution-time failures. However, in complex and contact-rich environments, geometric constraints, occlusions, and action-induced interactions may cause an otherwise reasonable plan to fail during execution. To address this challenge, **NeurVLA** introduces *Trajectory-Modeling Failure Prevention* ($P_{TM}$), which evaluates subsequent plans through explicit prediction and verification prior to execution.

Specifically, **NeurVLA** instantiates an executable program $p_p$ for future trajectory modeling and verification. The program $p_p$ first invokes ATM (Wen et al., 2023), a lightweight action-trajectory prediction model, to estimate the motion trajectories of task-relevant objects induced by the subsequent plan. The predicted trajectories are directly visualized on the current observation image, producing an augmented observation that explicitly represents the anticipated future evolution of the scene. Based on this augmented observation, $p_p$ then performs verification using the enhanced VP to assess whether the predicted trajectories would lead to potential failures. During execution, the intermediate variables of $p_p$ are organized as a dictionary and denoted as $d_p$. If a potential failure is detected by $p_p$, **NeurVLA** updates the plan and re-evaluates it through the same prediction-verification loop until a safe and feasible plan is identified and passed

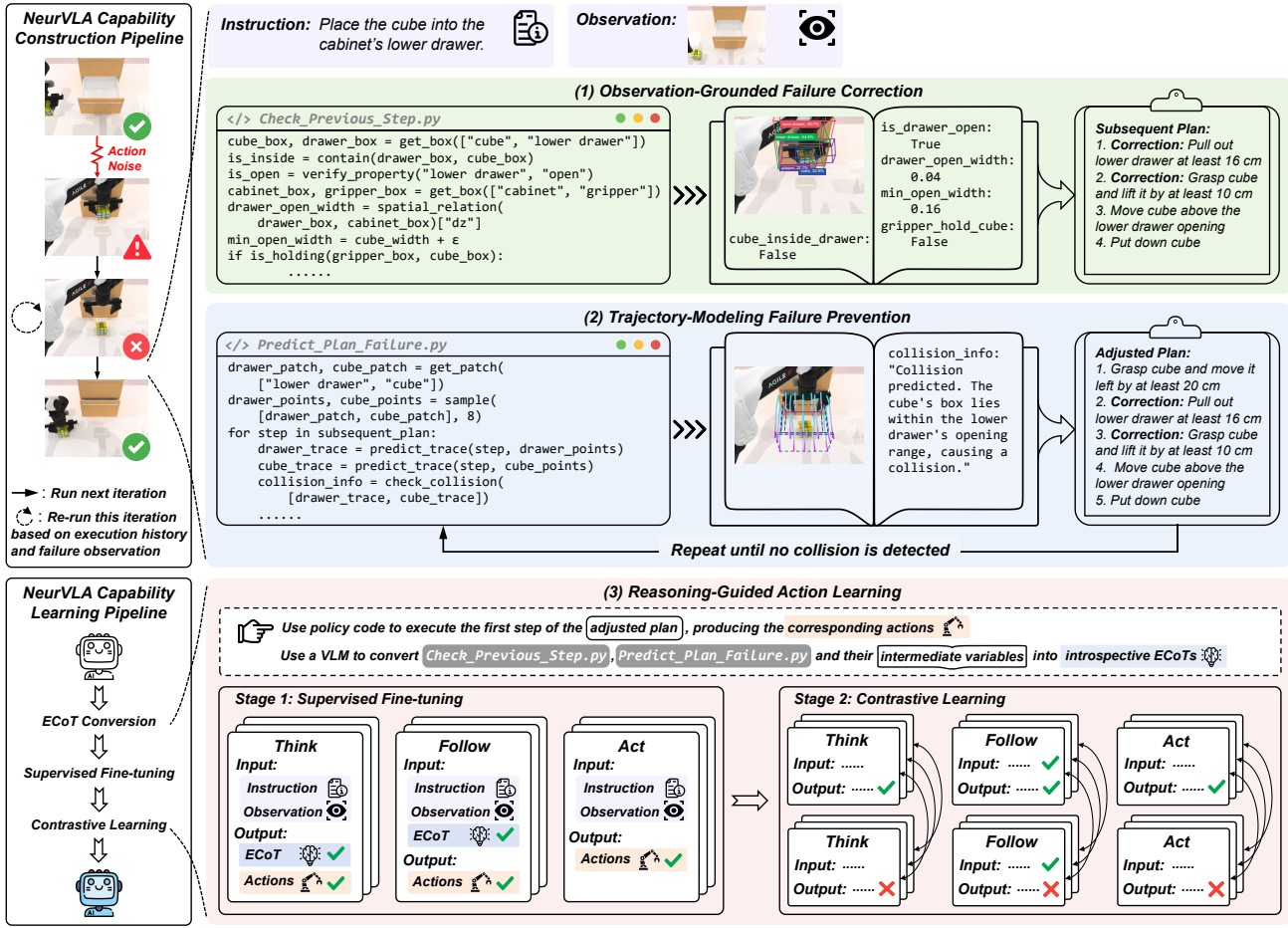

*Figure 2.* Overview of **NeurVLA**. **NeurVLA** jointly constructs and internalizes robust failure-handling capabilities for VLA models. It (1) programmatically checks completion of the previously executed actions from observations to enable fine-grained failure correction, (2) predicts potential failures in the subsequent plan via executable trajectory simulation for reliable failure prevention, and, after using policy code to obtain the corresponding actions, it (3) internalizes failure-handling capability by first converting the above reasoning processes into introspective ECoTs, and then performing SFT and contrastive learning on ECoT-actions.

to the execution stage. By grounding failure prevention in explicit and verifiable evidence, **NeurVLA** reliably aligns failure prediction with execution-time outcomes, reducing missed hazards and unnecessary conservative interventions in complex scenarios.

**Notably,** to support high-fidelity execution of fine-grained actions, **NeurVLA** then translates the first step of the verified subsequent plan into a policy code block that directly invokes parameterized control-primitive APIs and executes it in the environment. If execution fails, **NeurVLA** triggers two coordinated branches: **(i)** it proceeds to the next iteration with the failed observation, where $C_{OG}$ analyzes the failure based on post-execution observations and injects fine-grained corrective steps into subsequent plans; and **(ii)** it re-runs the current iteration by feeding back the execution history and failure observation, where $P_{TM}$ revises the plan to avoid repeating the same failure before execution.

### 3.3. Reasoning-Guided Action Learning

To internalize failure-handling capabilities in VLA models, **NeurVLA** introduces *Reasoning-Guided Action Learning ($L_{RG}$)*. Specifically, $L_{RG}$ first employs a VLM-based ECoT generator $G$ to convert the executed programs in $C_{OG}$ and $P_{TM}$ and their associated intermediate variables into an introspective ECoT $e$, i.e., $e = G(o, inst, p_c, d_c, p_p, d_p)$, yielding one ECoT-action instance per iteration and thus making the above reasoning processes learnable. This conversion is enforced by strict rule-based constraints via carefully designed prompts (see Appendix C) and is applied uniformly to both successful and failed iterations, thereby providing contrastive supervision signals. Leveraging these per-iteration ECoT-action instances, $L_{RG}$ then constructs training samples that consist of an observation, an instruction, an introspective ECoT, and the corresponding action, and represents them in three complementary training formats: Think, Act, and Follow. Given the observation and

*Table 1.* Comparison of VLA models on LIBERO.

| Method | Spatial | Object | Goal | Long | Avg |
|---|---|---|---|---|---|
| *Regression or classification-based VLA* | | | | | |
| OpenVLA | 84.7 | 88.4 | 79.2 | 53.7 | 76.5 |
| SpatialVLA | 88.2 | 89.9 | 78.6 | 55.5 | 78.1 |
| UniVLA | 96.5 | 96.8 | 95.6 | 92.0 | 95.2 |
| NORA | 92.2 | 95.4 | 89.4 | 74.6 | 87.9 |
|   - *CollabVLA* | 93.4 | 96.2 | 92.2 | 81.3 | 90.8 |
|   - *FailSafe* | 94.2 | 96.5 | 93.5 | 85.8 | 92.5 |
|   - **NeurVLA** | **97.1** | **98.3** | **96.7** | **92.2** | **96.1** |
| *Flow-matching or diffusion-based VLA* | | | | | |
| $\pi_0$ | 96.8 | 98.8 | 95.8 | 85.2 | 94.2 |
| ThinkAct | 88.3 | 91.4 | 87.1 | 70.9 | 84.4 |
| FPC-VLA | 87.0 | 92.0 | 86.2 | 82.2 | 86.9 |
| InstructVLA | 92.4 | 95.6 | 92.0 | 76.6 | 89.2 |
|   - *CollabVLA* | 93.6 | 96.4 | 93.2 | 80.7 | 91.0 |
|   - *FailSafe* | 93.9 | 96.5 | 93.6 | 81.1 | 91.3 |
|   - **NeurVLA** | **97.5** | **99.2** | **97.3** | **92.4** | **96.6** |

instruction, the Think format predicts both ECoTs and actions, the Act format predicts actions only, and the Follow format predicts actions conditioned on the observation, instruction, and a given ECoT.

Subsequently, $L_{RG}$ apply SFT using successfully executed samples across all three formats, enabling the model to internalize the reasoning patterns and action behaviors associated with successful execution. Benefiting from the two-branch design triggered upon execution failure in **NeurVLA**, each execution-failure sample is paired with a successfully executed sample obtained from the re-run iteration. Accordingly, $L_{RG}$ then leverage these paired samples across the three formats to perform contrastive learning, encouraging the model to prefer reasoning processes and actions that lead to successful execution while suppressing those associated with failures. Through this learning process, failure-handling capability is internalized as an intrinsic property of the VLA model. At deployment time, the model can be adapted to new environments by lightly fine-tuning using the Act format, since action realizations are environment-specific, while the learned reasoning capabilities for failure correction and prevention remain transferable across tasks and environments.

# 4. Experiment

We conduct comprehensive experiments to evaluate the effectiveness of **NeurVLA**. We begin with the experimental setup (Sec. 4.1), followed by results on simulated benchmarks (Sec. 4.2) and real-world task performance (Sec. 4.3). Sec. 4.4 provides ablation studies on capability construction

and learning components, along with evaluation of robustness (Sec. 4.5), failure correction and prevention (Sec. 4.6), and computational overhead (Sec. 4.7).

## 4.1. Experimental Setup

**Baselines.** We compare **NeurVLA** against two categories of representative VLA models with different action modeling paradigms. (1) Regression or classification-based VLA. This category includes OpenVLA (Kim et al., 2024), CoT-VLA (Zhao et al., 2025), SpatialVLA (Qu et al., 2025), and UniVLA (Bu et al., 2025). (2) Flow-matching or diffusion-based VLA. This category includes $\pi_0$ (Black et al., 2024), ThinkAct (Huang et al., 2025a), UniCoD (Zhang et al., 2025), and FPC-VLA (Yang et al., 2025b). In addition, we compare against representative failure-handling methods, including CollabVLA (Sun et al., 2025) and FailSafe (Lin et al., 2025). Detailed descriptions of all baseline models and methods are provided in the Appendix A.1.

**Benchmarks.** We evaluate **NeurVLA** on three widely used simulated manipulation benchmarks: LIBERO (Liu et al., 2023a), SimplerEnv (Li et al., 2024), and CALVIN (Mees et al., 2022). For CALVIN, we follow the standard ABC→D generalization protocol. Detailed descriptions of each benchmark are provided in the Appendix A.2.

**Model Setup.** To demonstrate the generality of our approach, we apply **NeurVLA** to two representative VLA models with fundamentally different action modeling paradigms: NORA (Hung et al., 2025) and InstructVLA (Yang et al., 2025a). NORA is a lightweight autoregressive VLA built on a vision-language model backbone, which predicts discretized action tokens via next-token prediction and can also generate textual outputs when trained with appropriate language supervision. InstructVLA, by contrast, couples autoregressive text generation with flow-matching-based action generation, where a flow model produces continuous actions conditioned on latent action representations derived from the generated text and visual context.

## 4.2. Main Results In Simulation

**LIBERO.** We evaluate **NeurVLA** on the LIBERO benchmark and compare it with representative VLA baselines across two action modeling paradigms. As shown in Table 1, **NeurVLA** improves the average success rate by 0.9 points over the strongest regression or classification-based baseline, and by 2.4 points over the strongest flow-matching or diffusion-based baseline. Notably, **NeurVLA** achieves a 23.0% relative improvement in average success rate compared to SpatialVLA. We attribute these improvements to **NeurVLA**'s explicit and verifiable 3D geometric reasoning, which enables metric-level state analysis beyond implicit spatial representations.

*Table 2.* Performance comparison on SimplerEnv.

| Method | Google Robot | | | | | | | | WidowX Robot | | | | | Avg |
|---|---|---|---|---|---|---|---|---|---|---|---|---|---|---|
| | Pick Coke Can | | Move Near | | Open/Close Drawer | | Avg | | Put Spoon | Put Carrot | Stack Blocks | Put Eggplant | Avg | |
| | VM | VA | VM | VA | VM | VA | VM | VA | | | Success | | | |
| *Regression or classification-based VLA* | | | | | | | | | | | | | | |
| OpenVLA | 16.3 | 54.5 | 46.2 | 47.7 | 35.6 | 17.7 | 32.7 | 40.0 | 0.0 | 0.0 | 0.0 | 4.1 | 1.0 | 24.6 |
| CoT-VLA | 60.3 | 71.8 | 61.5 | 57.2 | 48.0 | 35.2 | 56.6 | 54.7 | 17.9 | 18.2 | 15.1 | 42.6 | 23.5 | 44.9 |
| SpatialVLA | 81.0 | 89.5 | 69.6 | 71.7 | 59.3 | 36.2 | 70.0 | 65.8 | 20.8 | 20.8 | 25.0 | 70.8 | 34.4 | 56.7 |
| NORA | 81.6 | 56.2 | 69.0 | 77.9 | 30.1 | 24.5 | 60.2 | 52.9 | 71.3 | 40.9 | 35.7 | 83.8 | 57.9 | 57.0 |
| - *CollabVLA* | 85.8 | 69.1 | 77.8 | **86.1** | 42.8 | 32.4 | 68.8 | 62.5 | 76.5 | 49.2 | 46.1 | 85.9 | 64.4 | 65.2 |
| - **NeurVLA** | **90.6** | **89.8** | **82.1** | 84.5 | **63.4** | **41.7** | **78.7** | **72.0** | **82.7** | **56.3** | **52.8** | **89.9** | **70.4** | **73.7** |
| *Flow-matching or diffusion-based VLA* | | | | | | | | | | | | | | |
| $\pi_0$ | 72.7 | 75.2 | 65.3 | 63.7 | 38.3 | 25.6 | 58.8 | 54.8 | 29.1 | 0.0 | 16.7 | 62.5 | 27.1 | 46.9 |
| ThinkAct | 92.0 | 84.0 | 72.4 | 63.8 | 50.0 | 47.6 | 71.5 | 65.1 | 58.3 | 37.5 | 8.7 | 70.8 | 43.8 | 60.1 |
| FPC-VLA | 95.3 | 91.3 | 93.8 | 86.7 | **76.4** | 30.7 | 88.5 | 69.6 | 79.2 | 58.3 | 45.8 | 75.0 | 64.6 | 74.2 |
| InstructVLA | 79.6 | 92.3 | 68.3 | 71.9 | 52.3 | 61.7 | 66.7 | 75.3 | 47.1 | 40.4 | 20.6 | 71.5 | 44.9 | 62.3 |
| - *CollabVLA* | 86.1 | 93.6 | 80.4 | 82.7 | 63.2 | 65.3 | 76.6 | 80.5 | 62.2 | 49.7 | 31.9 | 76.3 | 55.0 | 70.7 |
| - **NeurVLA** | **96.5** | **98.2** | **94.8** | **95.1** | 74.9 | **76.6** | **88.7** | **90.0** | **83.8** | **63.5** | **52.8** | **91.2** | **72.8** | **83.8** |

*Table 3.* Evaluation on the CALVIN ABC→D benchmark.

| Method | Tasks completed in a row (%) ↑ | | | | | Avg Len. |
|---|---|---|---|---|---|---|
| | 1 | 2 | 3 | 4 | 5 | |
| *Regression or classification-based VLA* | | | | | | |
| OpenVLA | 91.3 | 77.8 | 62.0 | 52.1 | 43.5 | 3.27 |
| UniVLA | 95.5 | 85.8 | 75.4 | 66.9 | 56.5 | 3.80 |
| NORA | 93.9 | 82.7 | 70.2 | 61.4 | 51.4 | 3.60 |
| - *FailSafe* | 95.6 | 87.1 | 78.3 | 72.5 | 64.4 | 3.98 |
| - **NeurVLA** | **96.4** | **89.0** | **82.1** | **74.8** | **66.3** | **4.09** |
| *Flow-matching or diffusion-based VLA* | | | | | | |
| $\pi_0$ | 93.8 | 85.0 | 76.7 | 68.1 | 59.9 | 3.84 |
| UniCoD | 97.3 | 89.5 | 82.3 | 75.2 | 67.0 | 4.11 |
| InstructVLA | 93.1 | 83.8 | 72.6 | 63.6 | 55.4 | 3.69 |
| - *FailSafe* | 95.4 | 87.9 | 77.8 | 70.9 | 63.8 | 3.96 |
| - **NeurVLA** | **97.9** | **90.3** | **83.4** | **75.8** | **69.6** | **4.17** |

**SimplerEnv.** As shown in Table 2, **NeurVLA** consistently achieves the strongest average performance on both Google Robot and WidowX Robot. In the regression or classification-based setting, **NeurVLA** improves the average success rate by approximately 10.9% on Google Robot and 9.3% on WidowX Robot over the strongest baselines. In the flow-matching or diffusion-based setting, **NeurVLA**

further improves the average success rate by about 6.01% on Google Robot and 12.7% on WidowX Robot compared to the respective strongest baselines. Notably, **NeurVLA** consistently outperforms CollabVLA across both robots and paradigms. We attribute these improvements to its more reliable failure prevention grounded in explicit future motion-trajectory modeling and geometric hazard checking, rather than purely semantic self-reflection.

**CALVIN.** As shown in Table 3, **NeurVLA** consistently completes longer task sequences with higher success rates than other baselines. In the regression or classification-based setting, **NeurVLA** enhances the NORA backbone with a 13.6% relative increase in average completion length, surpassing the strongest baseline, FailSafe, by approximately 2.8%. In the flow-matching or diffusion-based setting, **NeurVLA** boosts the InstructVLA backbone's average completion length by 13.0%, yielding an improvement of about 1.5% over the strongest baseline, UniCoD. Notably, **NeurVLA** consistently outperforms FailSafe across both paradigms. We attribute these improvements to internalizing failure-handling capability directly into the VLA model rather than relying on externally invoked recovery mechanisms, which leads to more robust and generalizable behavior in unseen tasks and environments.

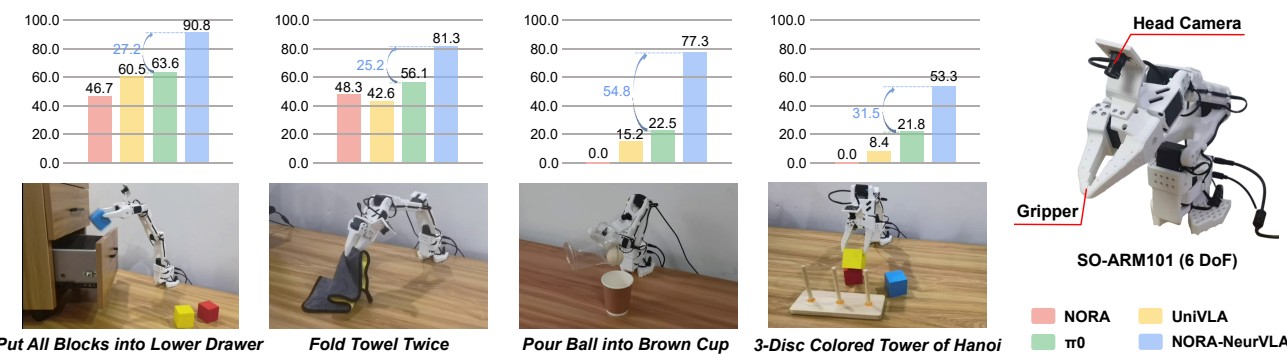

*Figure 3.* Comparison of NORA, UniVLA, $\pi_0$, and NORA with **NeurVLA** on four real-world tasks.

## 4.3. Real-World Experiment

As shown in Figure 3, we evaluate **NeurVLA** across four challenging real-world tasks: *Put All Blocks into Lower Drawer*, *Fold Towel Twice*, *Pour Ball into Brown Cup*, and *3-Disc Colored Tower of Hanoi*, using the SO-ARM101 robotic arm for the experiments. The results demonstrate that **NeurVLA** achieves state-of-the-art performance across all tasks, improving the average success rate by 51.9 points over the backbone model NORA and surpassing the strongest baseline $\pi_0$ by 34.7 points. Notably, in tasks that require long-horizon, fine-grained actions and complex reasoning, such as *Pour Ball into Brown Cup* and *3-Disc Colored Tower of Hanoi*, other baselines achieve success rates of less than 25%, while **NeurVLA** maintains over 50% success. We attribute this improvement to **NeurVLA**'s internalized fine-grained failure correction, reliable failure prevention, and the reasoning capabilities enabled by ECoT, allowing it to be reliably deployed in dynamic and spatially complex real-world scenarios.

## 4.4. Ablation Study

To better understand the contributions of different components in **NeurVLA**, we conduct ablation studies on both capability construction and capability learning on SimplerEnv-WidowX, as summarized in Table 4. **(1) Impact of failure-handling capability construction components.** Introducing $C_{OG}$ yields the largest individual improvement, corresponding to a 23.4% relative increase over the backbone model, while incorporating $P_{TM}$ alone results in a 17.6% relative improvement. When $C_{OG}$ and $P_{TM}$ are combined, the relative performance gain further rises to 44.1%, exceeding the sum of their individual improvements. This result reveals a complementary and mutually reinforcing effect between the two construction components. **(2) Impact of failure-handling capability learning components.** Replacing SFT with $L_{RG}$ yields an additional 12.5% relative improvement. Integrating both programmatic capability construction and reasoning-guided capability internalization achieves the strongest improvement, highlighting the neces-

*Table 4.* Ablation study on construction and learning components.

| | Method | Construct | | Learn | | SimplerEnv-WidowX |
|---|---|---|---|---|---|---|
| | | $C_{OG}$ | $P_{TM}$ | SFT | $L_{RG}$ | |
| 0 | Backbone | | | ✓ | | 44.9 |
| 1 | + $C_{OG}$ | ✓ | | ✓ | | 55.4 |
| 2 | + $P_{TM}$ | | ✓ | ✓ | | 52.8 |
| 3 | + $C_{OG}$ + $P_{TM}$ | ✓ | ✓ | ✓ | | 64.7 |
| 4 | + $C_{OG}$ + $L_{RG}$ | ✓ | | | ✓ | 62.5 |
| 5 | + $P_{TM}$ + $L_{RG}$ | | ✓ | | ✓ | 59.1 |
| 6 | **InstructVLA** *with* **NeurVLA** | ✓ | ✓ | | ✓ | **72.8** |

sity of jointly constructing and internalizing failure-handling capability for robust and generalizable performance. **In addition**, we compare ECoTs derived from programs used in $C_{OG}$ and $P_{TM}$ with those directly generated by an LLM for failure correction and prevention, and analyze their impact on model performance, as detailed in Appendix B.

## 4.5. Robustness Evaluation

We evaluate the robustness of InstructVLA with **NeurVLA** under various perturbations in both simulated and real-world environments. For simulation, as shown in Figure 4a, **NeurVLA** achieves the highest success rates across most perturbations. In action and environmental uncertainties, **NeurVLA** improves the success rate by 42.4% and 49.5% compared to the strongest baseline. This improvement can be attributed to $C_{OG}$, which enables rapid recovery from transient action disturbances, and $P_{TM}$, which proactively avoids failure-inducing interactions under dynamic environmental changes. For instruction uncertainties, benefiting from the ECoT learned in $L_{RG}$, **NeurVLA** further enhances the backbone's already high success rate. Regarding observation uncertainties, **NeurVLA** maintains performance despite imperfect visual evidence, while performance drops more under image shifts and rotations, which distort spatial

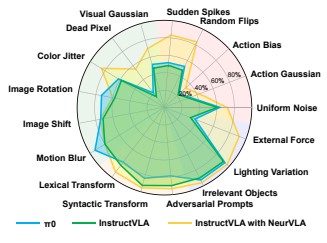
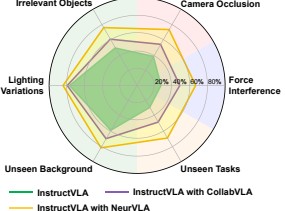

(a) Performance under various simulated perturbations.

(b) Performance under various real-world perturbations.

Figure 4. Robustness of **NeurVLA** across different perturbations in both simulated and real-world environments.

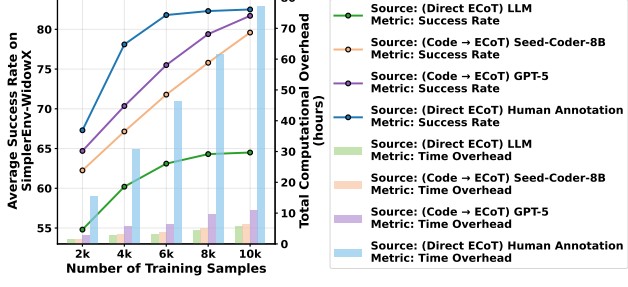

Figure 6. Overhead assessment of different ECoT sources under success-rate and computation-overhead metrics.

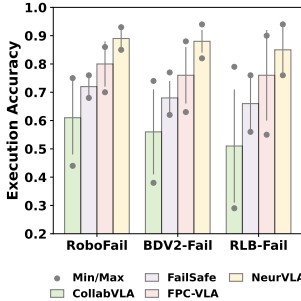
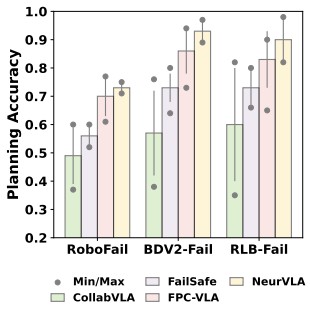

(a) Execution accuracy of different failure-handling methods on three failure benchmarks.

(b) Planning accuracy of different failure-handling methods on three failure benchmarks.

Figure 5. Evaluation results of failure correction and prevention.

configurations and mismatch coordinate-based observation-grounded checks, further degrading performance. For the real world, as shown in Figure 4b, **NeurVLA** yields consistent gains under unseen background, lighting variations, irrelevant objects, force interference, and camera occlusion, aligning with its strong performance in simulations. Moreover, the unseen tasks highlights **NeurVLA**'s superior ability to generalize to real-world scenarios.

### 4.6. Evaluation of Failure Correction and Prevention

As shown in Figure 5, we evaluate four failure-handling methods on three failure benchmarks (Pacaud et al., 2025), reporting execution and planning binary accuracies. The former judges whether each step succeeds based on pre/post observations, while the latter judges whether a proposed plan is correct given the task and initial scene. Across all three benchmarks, **NeurVLA** achieves the best accuracies for both execution and planning and also shows smaller variability. Since execution detection enables timely correction during rollout and planning detection anticipates risks before execution, **NeurVLA**'s strong performance indicates more reliable failure correction and prevention.

### 4.7. Overhead Assessment

**Experimental Setup.** As shown in Figure 6, we conduct an overhead assessment under four settings: *(Direct ECoT) LLM* follows ECoT (Zawalski et al., 2024) and uses GPT-5 to directly generate natural-language ECoTs from observation and instruction; *(Code→ECoT) Seed-Coder-8B* and *(Code→ECoT) GPT-5* follow **NeurVLA**'s capability construction pipeline by generating executable code and converting the resulting execution traces into ECoTs; and *(Direct ECoT) Human Annotation* uses human-annotated ECoTs. For a fair comparison, all settings use the same SFT protocol for training.

**Experimental Results and Analysis.** **(1) Favorable overhead-performance trade-off.** Compared with *(Direct ECoT) LLM*, *(Code→ECoT) GPT-5* consistently achieves substantially higher success rates by roughly 10–18 points with only $\sim 2\times$ time overhead, indicating that grounding ECoTs in execution traces yields stronger supervision than directly generating ECoTs. While *(Direct ECoT) Human Annotation* attains the best success rates, it incurs much higher overhead, roughly 8–10× that of *(Code→ECoT) GPT-5*, and its advantage becomes less pronounced as the number of training samples increases. Overall, **NeurVLA** delivers the most favorable overhead-performance trade-off. **(2) Limited overhead with smaller models.** Replacing the code and ECoT generator from GPT-5 with smaller open-source Seed-Coder-8B (Seed et al., 2025) results in only a minor success-rate drop of about 3–6 points, while reducing the time overhead by roughly half. This demonstrates that **NeurVLA** can substantially lower overhead by leveraging efficient smaller open-source code models, with negligible performance degradation in practice. **(3) Efficient overhead-to-gain scaling.** For *(Code→ECoT)*, increasing the number of training samples from 2k to 10k yields large and steady gains, indicating that additional overhead is consistently translated into meaningful performance improvements. In contrast, *(Direct ECoT)* settings exhibit diminishing returns as data scales, highlighting **NeurVLA**'s more efficient overhead-to-performance conversion.

## 5. Conclusion

In this work, we present **NeurVLA**, a neural-symbolic framework unleashing failure-handling capability of VLA models. It integrates *Observation-Grounded Failure Correction* and *Trajectory-Modeling Failure Prevention* to ensure fine-grained, reliable failure handling and further introduces *Reasoning-Guided Action Learning*, enabling VLA model to internalize failure-handling capabilities. Experiments in simulation and real-world validate its effectiveness.

**Acknowledgement.** This work was supported by the National Key Research and Development Program of China (2025ZD0123100), National Natural Science Foundation of China (62436007), Key R&D Program of Zhejiang (2026SDXT005), Zhejiang NSF (LQK26F020001).

## Impact Statement

This work aims to advance robust and reliable vision-language-action models for embodied robotic manipulation. By improving failure correction and failure prevention through programmatic reasoning and reasoning-guided action learning, **NeurVLA** may benefit robotic systems deployed in assistive manipulation, logistics, household service, and industrial automation, where recovering from execution errors and anticipating unsafe motions are important for reducing damage and improving operational reliability. At the same time, improved autonomy in physical agents may introduce risks if such systems are deployed without appropriate safeguards, including erroneous actions caused by perception or prediction failures, over-reliance on simulated evaluation, distribution shifts in real environments, privacy concerns from visual sensing, and potential misuse in applications that could harm people or property. Therefore, systems based on this work should be used with task-specific risk assessment, human oversight in safety-critical settings, physical constraints and emergency-stop mechanisms, careful validation under deployment conditions, and compliance with relevant regulations and institutional safety protocols. We do not intend this work to remove the need for human supervision or formal safety verification; rather, it should be viewed as a step toward more interpretable and failure-aware embodied learning systems.

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

# Appendix

## A. Details of Experimental Setup

### A.1. Baseline Details

We compare `NeurVLA` against two categories of representative VLA models with different action modeling paradigms, together with representative failure-handling methods.

***Regression or classification-based VLA:***

- OpenVLA (Kim et al., 2024): It is an autoregressive, classification-based VLA model that discretizes continuous robot actions into tokens and predicts them using a large pretrained vision-language model. It is trained on large-scale real-world robot demonstrations and supports efficient fine-tuning for generalist control across diverse tasks and robot embodiments.

- CoT-VLA (Zhao et al., 2025): It is an autoregressive VLA model with explicit visual CoT reasoning, which predicts future visual goal frames before generating short action sequences to reach these goals. It incorporates intermediate visual reasoning into action generation to improve performance on complex manipulation tasks requiring temporal planning.

- SpatialVLA (Qu et al., 2025): It is an autoregressive, classification-based VLA model that emphasizes spatial understanding by injecting ego-centric 3D information into visual observations and representing robot actions with adaptive discretized spatial action grids. It is pre-trained on large-scale real-world robot episodes to learn generalizable spatial action representations and supports zero-shot deployment and efficient adaptation across different robot embodiments.

- UniVLA (Bu et al., 2025): It is an autoregressive, classification-based VLA model designed for cross-embodiment generalization, which derives task-centric latent action representations from large-scale video data under language conditioning. By learning and decoding latent actions across heterogeneous embodiments, it enables scalable and efficient transfer of policies to diverse robots and environments.

***Flow-matching or diffusion-based VLA:***

- $\pi_0$ (Black et al., 2024): It is a flow-based VLA model that builds a flow-matching architecture on top of a pretrained vision-language model to generate continuous robot actions. It is trained on large-scale, multi-embodiment robot datasets and supports zero-shot execution, language-conditioned control, and efficient fine-tuning across diverse dexterous manipulation tasks.

- ThinkAct (Huang et al., 2025a): It is a diffusion-based VLA framework that couples high-level embodied reasoning with low-level action execution by conditioning an action generation model on compact visual plan latents. It trains a multimodal LLM to produce reinforced reasoning plans, which guide downstream action execution for long-horizon planning and adaptive behavior in complex environments.

- UniCoD (Zhang et al., 2025): It is a flow-based VLA model that learns predictive visual representations through large-scale pretraining on instructional manipulation videos and maps these representations to robot actions during fine-tuning. By modeling future visual dynamics and leveraging continuous predictive representations, it enables robust policy learning across simulation and real-world out-of-distribution tasks.

- FPC-VLA (Yang et al., 2025b): It is a diffusion-based VLA model that integrates a supervisory module for failure prediction and correction via vision-language queries. The supervisor evaluates action viability and provides corrective guidance when risks are detected, while a dual-stream fusion mechanism refines actions using past predictions.

*Failure-handling methods*:

- CollabVLA (Sun et al., 2025): It is a failure-handling method that integrates VLM-based self-reflective reasoning with diffusion-based action generation to support collaborative and assistive robot behavior. It produces explicit reflection signals and solicits human guidance under uncertainty or repeated failure through a two-stage training process combining action grounding and reflection tuning.
- FailSafe (Lin et al., 2025): It is a failure-handling method that focuses on scalable generation of failure-recovery data by automatically pairing diverse failure cases with executable recovery actions in simulation. It is applied as an external failure detection and recovery system to enhance existing VLA models by fine-tuning a separate vision-language model on the generated failure data.

### A.2. Benchmark Details

We evaluate **NeurVLA** on the following simulated benchmarks:

- LIBERO (Liu et al., 2023a): It is a simulated lifelong robot manipulation benchmark that procedurally generates language-conditioned tasks grouped into four suites (Spatial, Object, Goal, and Long) to study knowledge transfer under controlled distribution shifts, and it provides high-quality demonstrations to support sample-efficient learning.
- SimplerEnv (Li et al., 2024): It is a real-to-sim evaluation suite designed to assess transferability and generalization of robot policies trained on real-world video data under controlled visual distribution shifts. It evaluates policies on the WidowX and Google Robot platforms with variations in lighting, textures, object colors, and camera viewpoints, covering representative manipulation tasks.
- CALVIN (Mees et al., 2022): It is a simulated benchmark for long-horizon, language-conditioned robot manipulation, where agents execute compositional multi-step behaviors from sequential instructions. It spans four environments (A, B, C, and D) with 34 tasks and 1,000 language instructions.

## B. Additional Ablations: Program-Derived vs. LLM-Generated ECoTs

**Experimental Setup.** We compare **NeurVLA** with a variant that replaces program-derived ECoTs with ECoTs directly generated by an LLM for failure correction and prevention. Specifically, this variant generates ECoTs from the observation and instruction only, without conditioning on program execution traces or verification signals produced by our *Observation-Grounded Failure Correction* and *Trajectory-Modeling Failure Prevention*. For a fair comparison, we keep the backbone VLA model, training protocol, execution interface, and the budgets for failure correction and failure prevention identical.

*Table 5.* Ablations: replacing program-derived ECoTs with LLM-direct ECoTs for failure correction and prevention.

| | Method | Correction ECoT Source | | Prevention ECoT Source | | SimplerEnv-WidowX |
|---|---|---|---|---|---|---|
| | | Program-derived | LLM-direct | Program-derived | LLM-direct | |
| 0 | InstructVLA with **NeurVLA** | ✓ | | ✓ | | 72.8 |
| 1 | w/o Program-derived Correction ECoT | | ✓ | ✓ | | 57.2 |
| 2 | w/o Program-derived Prevention ECoT | ✓ | | | ✓ | 62.5 |
| 3 | w/o Program-derived ECoTs (both) | | ✓ | | ✓ | 50.4 |
| 4 | InstructVLA (Backbone) | | | | | 44.9 |

**LLM-direct:** ECoTs are directly generated by an LLM from the observation and instruction only, without conditioning on program execution traces or verification signals.

**Experimental Results and Analysis.** As shown in Table 5, for ECoTs used in failure correction, replacing the ECoTs derived from programs with those directly generated by an LLM results in a 21.4% relative drop in success rate, whereas for

ECoTs used in failure prevention, the same replacement leads to a 14.1% relative drop. When both correction and prevention ECoTs are replaced by LLM-direct ECoTs simultaneously, the success rate further decreases by 30.8%, but still remains 12.2% higher than the InstructVLA backbone. Overall, these results confirm that incorporating ECoTs is crucial for effective failure handling, with program-derived ECoTs consistently delivering stronger gains than LLM-direct natural-language ECoTs, and further suggest that correction-oriented ECoTs have a larger impact on overall success than prevention-oriented ones in this setting.

## C. Details of ECoT Conversion Prompts

As shown in Figure 7, we provide details of introspective ECoT conversion prompt.

---

**Prompt for Introspective ECoT Conversion**

**# INSTRUCTION #**
You will be given two complete executable programs used in Vision-Language-Action Models, one for Failure Correction and one for Failure Prevention, along with the intermediate variable values produced during their actual execution. Your task is to (1) infer the true executed paths and final outcomes of both programs using the intermediate values, and then (2) convert the combined results into a single introspective Embodied Chain of Thought (ECoT) that can guide the robot's next action.

**# FAITHFULNESS RULES #**
(1) Input faithful: rely strictly on the program text and the intermediate variable values; do not add any assumptions or imagined scene details.
(2) Path faithful: if the code contains branch(es) or iteration(s), describe only the branch or iterations that were actually executed in this run; never mention unexecuted paths.
(3) Entity faithful: mention only objects, locations, states, and quantities that appear in the intermediate variables; do not invent new entities, relations, or actions.
(4) Number faithful: if distances, angles, thresholds, or displacements are explicitly provided, you may use them; if not provided, use only relative phrasing and never fabricate numbers.
(5) Outcome faithful: the correction conclusion must match the Failure Correction program's final decision and recommendation; the prevention conclusion must match the Failure Prevention program's risk assessment and the verified adjustment (if any). Never turn uncertain or unverified into done or safe.
(6) Insufficient information: if key intermediate variables are missing such that the executed path or final outcomes cannot be determined faithfully, output only: "We lack sufficient information to faithfully describe this reasoning step."

**# LANGUAGE & FORMATTING RULES #**
(1) Human like introspection: use first person plural (we), concise and natural. No opinions, no extra explanations, no analysis of significance.
(2) Single introspective ECoT: output exactly one introspective ECoT in natural language that covers both correction (if it happened) and prevention (if it happened), and ends with what we will do next (the immediate adjustment or continuation).
(3) Do not mention code: do not reference variable names, function or method names, operators, types, line numbers, file names, or say the program or variable indicates or returns.
(4) Your response should only contain the final natural language description with no additional explanations or code.

**# CONVERSION PROCEDURE #**
Step A: Reconstruct Failure Correction
A1. Determine whether the previous step was incomplete or failed based on intermediate variables.
A2. If correction is needed: identify the specific unmet condition (for example misalignment, insufficient distance, not reaching target state) and the computed adjustment (if provided).
A3. Produce precise correction outcome: what we must do first to fix the current state (or record no correction needed).
Step B: Reconstruct Failure Prevention
B1. Determine whether a future failure or risk (for example collision, obstruction, infeasibility) is predicted based on intermediate variables.
B2. If prevention is needed: identify the risk type and its trigger condition (as represented in intermediate variables), plus the avoidance or plan adjustment and any verification result (if provided).
B3. Produce the prevention outcome: the updated plan that is verified as safer or executable (or record no plan adjustment needed).
Step C: Fuse into One Introspective ECoT (Describe only what actually happened)
C1. If correction is needed, first state we found the current outcome is not achieved and what we will adjust immediately (use numbers only if provided).
C2. If prevention is needed, state we predict a risk and how we will adjust to avoid it (use numbers only if provided), referencing only what intermediate variables support.
C3. If neither is needed, state we confirmed the step is achieved and no obvious risk is predicted, so we proceed with the next action.

**# EXAMPLES #**
[EXAMPLES]

**# INPUT #**
Failure Correction Program: [Program]
Failure Correction Intermediate Variables: [Values of intermediate variables]
Failure Prevention Program: [Program]
Failure Prevention Intermediate Variables: [Values of intermediate variables]

**# OUTPUT #**

---

*Figure 7.* Prompt for Introspective ECoT Conversion.

