# OpenReview forum: "NeurVLA: Unleashing Failure-Handling Capability of Vision-Language-Action Models via Neural-Symbolic Reasoning"
_ICML.cc/2026/Conference — ICML 2026 regular_

### Official Review · Reviewer_n1Yt · 2026-03-02

**Soundness:** 3
**Presentation:** 3
**Significance:** 3
**Originality:** 3
**Overall Recommendation:** 4
**Confidence:** 3

**Summary:**

The paper proposes a neural-symbolic framework unleashing failure-handling capability of VLA models.

**Compliance With Llm Reviewing Policy:**

Affirmed.

**Key Questions For Authors:**

1. The paper refers to “coarse-grained adjustment” in previous work, but this term is not sufficiently explained. It would be helpful to provide a clearer definition or brief clarification to help readers better understand its meaning and role in the context of the discussion.

**Limitations:**

Refer above

**Strengths And Weaknesses:**

- Strengths:
The paper proposes a method aimed at alleviating failure occurrences and improving failure recovery, which are critical challenges in robotic manipulation. The experimental evaluation is conducted across multiple simulators, and the real-world tasks involve long-horizon scenarios, making them well-suited to assess the effectiveness and robustness of the proposed approach.

- Weaknesses:
1. The method relies on several external models, which may potentially affect inference efficiency. It would be helpful for the authors to report the inference speed of the proposed method and analyze whether the additional components introduce significant computational overhead.
2. The term “neural-symbolic” is used throughout the paper; however, it is not clear to me how the proposed approach qualifies as neural-symbolic.

---

> ### Author Rebuttal · Authors · 2026-03-31
>
> **W1 (Inference efficiency and overhead):**
>
> We clarify this point from three aspects:
>
> - **NeurVLA is a two-stage framework for introspective ECoT construction and VLA learning.** It first generates and executes programs for failure correction and prevention to construct introspective ECoTs, and then uses these ECoTs to train the VLA to internalize failure-handling capabilities.
> - **The external models in NeurVLA are used only for ECoT construction.** The external models, including 3D-MOOD and ATM, are used during ECoT construction rather than online deployment. Inspired by HyT [1], we design Reasoning-Guided Action Learning to internalize the failure-handling capabilities and specialized knowledge from these external models embedded in these ECoTs into the VLA.
> - **NeurVLA remains efficient at deployment.** At deployment time, NeurVLA directly outputs actions without invoking the external program-generation and execution pipeline online.
>
> **Sec. 4.7** of main text compares our ECoT construction pipeline with other ECoT construction methods in terms of performance-overhead trade-off, as shown below.
>
> |# Samples|Direct ECoT(MLLM) SR|Direct ECoT(MLLM) Time(h)|Code→ECoT(Seed-Coder-8B) SR|Code→ECoT(Seed-Coder-8B) Time(h)|Code→ECoT(GPT-5) SR|Code→ECoT(GPT-5) Time(h)|Direct ECoT(Human) SR|Direct ECoT(Human) Time(h)|
> |-|-|-|-|-|-|-|-|-|
> |2k|54.8|1.2|62.3|1.2|64.7|2.1|67.3|15.4|
> |4k|60.2|2.3|67.2|2.5|70.4|4.3|78.1|30.8|
> |6k|63.1|3.5|71.8|3.8|75.5|6.5|81.8|46.2|
> |8k|64.3|4.6|75.8|5.1|79.4|8.7|82.3|61.6|
> |10k|64.5|5.7|79.6|6.4|81.7|10.8|82.5|77.0|
>
> Here, we further compare NeurVLA with both UniVLA and π0.5 on SimplerEnv and in real-world settings in terms of inference frequency and task performance, as shown below.
>
> |Method|SimplerEnv Inference Freq.(Hz)|Google Robot Avg. VM|Google Robot Avg. VA|WidowX Avg. Grasp SR|WidowX Avg. Task SR|Real-World Inference Freq.(Hz)|Real-World Task SR|
> |-|-|-|-|-|-|-|-|
> |UniVLA|12.1|39.2|49.4|71.5|48.0|10.2|32.8|
> |π0.5|15.0|80.3|81.1|90.3|65.9|12.5|70.5|
> |NeurVLA|11.8|88.7|90.0|94.8|70.4|9.9|77.9|
>
> The results show that:
> - **NeurVLA achieves a favorable performance-overhead trade-off in ECoT construction.** Compared with direct generation, NeurVLA attains stronger performance with more efficient overhead-to-performance conversion, while remaining far less costly than human annotation. This advantage also persists across data scales and when using smaller open-source models.
> - **NeurVLA remains efficient and effective at deployment.** Its inference frequency is comparable to UniVLA and only moderately lower than π0.5, while attaining substantially stronger performance, outperforming UniVLA by 75.1% and π0.5 by 8.7% across simulation and real-world evaluations.
>
> ------
>
> **W2 (Clarification of neural-symbolic claim)**:
>
> We thank the reviewer for this important comment. Specifically:
>
> - **On the intended meaning part,** we agree that "neural-symbolic" may be too strong if interpreted in the classical symbolic-logic sense. Our intended emphasis is an executable, programmatic, and coordinate-grounded reasoning framework that combines neural modules with structured visual-program execution.
> - **On the symbolic/programmatic part,** what we intend to emphasize is the use of executable visual programs with explicit variables, operators, and verifiable conditions for structured reasoning over observations and geometric states. The variables include task-relevant objects, gripper/object states, predicted trajectories, and intermediate variables, while the primitives include containment checks, spatial relation reasoning, metric distance computation, trajectory prediction, and collision checking. Rather than relying on formal axioms or theorem proving, the programs are grounded by explicit operators and geometric constraints in a coordinate system.
> - **To avoid ambiguity, we will revise the wording in the next version to better reflect this programmatic nature.**
>
> ------
>
> **Q1 (Coarse-grained adjustment)**:
>
> - **For coarse-grained adjustment,** it refers to failure correction methods such as FPC-VLA that produce discretized direction/magnitude adjustments without precise localization or distance measurement in the current scene. Such adjustments are weakly grounded in the actual spatial configuration, often leave residual deviations after correction, and may lead to accumulated errors and repeated failures over subsequent steps.
> - **For fine-grained adjustment in NeurVLA,** it refers to corrective steps that are precisely grounded in the current observation and geometric state, making failure correction accurate, grounded, robust, and generalizable.
> - **For examples,** coarse-grained adjustment corresponds to corrective steps such as "move slightly left" or "move forward more", whereas fine-grained adjustment corresponds to more precise corrective steps such as "move 5 cm to the left" or "move 20 cm forward".
>
> [1] Mazzaglia P, et al. Hybrid training for vision-language-action models. 2025.

---

> > ### Author Rebuttal · Reviewer_n1Yt · 2026-04-03
> >
> > The authors made a dedicated effort to address the issues I raised, which I greatly appreciate. The rebuttal explains my concerns from multiple perspectives and is very clear. I will maintain my score and am inclined to accept the paper.

---

> > > ### Author Response · Authors · 2026-04-03
> > >
> > > Thank you very much for reviewing our response. We greatly appreciate your valuable comments and feedback.

---

### Official Review · Reviewer_Tfgc · 2026-03-08

**Soundness:** 3
**Presentation:** 3
**Significance:** 3
**Originality:** 2
**Overall Recommendation:** 4
**Confidence:** 4

**Summary:**

In this paper, the authors study the problems of failure detection and correction in vision-language-action (VLA) models. To this end, the authors propose a "neural-symbolic" framework wherein a VLA generates code to check/verify the status of the previous execution step, makes a plan accordingly, and generates another code to verify if the generated trajectory is reasonable. Across different benchmarks, the proposal VLA model is shown to provide improvements over the baseline and compared models.

**Compliance With Llm Reviewing Policy:**

Affirmed.

**Final Justification:**

The authors have addressed almost all of my concerns. The remaining bits: (1) The method is computationally more expensive compared to a VLA. (2) The approach introduces other challenges with the correctness of the generated code and the complexities of using external tools. (3) The neuro-symbolic claim, which has been admitted to be over-claimed by the authors, needs to be rephrased throughout the text.

I have increased my recommendation.

**Key Questions For Authors:**

Please see Weaknesses.

**Limitations:**

The authors have NOT included any discussion on the limitations and potential negative societal impact of their VLA model.

**Strengths And Weaknesses:**

Strengths:
+ Failure detection and correction are important problems in VLAs and ML.
+ The propose approach appears to be novel.
+ Significant performance gains are reported.

Weaknesses:

1. I have concerns about the practicality of the proposed approach.

1.1. The paper provides some analysis on the computational overhead of the proposed framework in comparison to a LLM using COT. However, the comparison should have been made with the VLAs that were compared in terms of success rate.

1.2. The computational overhead of the proposed VLA is prohibitive to be used in practice on a robot.

1.3. What prevents the system from entering an infinite loop?

1.4. "it proceeds to the next iteration with the failed observation, where COG analyzes the failure based on post-execution observations and injects fine-grained corrective steps into subsequent plans;" => What if the failure is more global, requiring more global correction?

1.5. The VLA generates code for verifying different steps. However, it is not clear how reliable the generated codes are and how the codes themselves should be verified.

2. Not sure about the neural-symbolic claim of the framework.

2.1. "neural-symbolic paradigm" implies that the paper uses a symbolic language/program together with the neural architecture. However, the framework just generates Python code, which is executed. Yes, Python code is symbolic; however, it is not a "symbolic" language/program in the sense of the "neural-symbolic paradigm".

2.2. "In each iteration, NeurVLA instantiates an executable neural-symbolic program pc based on the enhanced VP, and uses it to analyze the current observation to assess whether the previous action has been successfully completed." => If you argue for a symbolic program, what are the variables, axioms, predicates of this symbolic program?

2.3. "NeurVLA constructs an enhanced version of visual programming (VP) (Gupta & Kembhavi, 2023) by augmenting standard VP with geometry functions for spatial relation reasoning and metric distance computation, defined in the same coordinate system as Omni3D" => Isn't it better to call it programmatic-VLA or something similar?

3. Many details are not sufficiently explained.

3.1. "The predicted trajectories are directly visualized on the current observation image, producing an augmented observation that explicitly represents the anticipated future evolution of the scene. Based on this augmented observation, pp then performs verification using the enhanced VP to assess whether the predicted trajectories would lead to potential failures." => How can failure be detected from the image projection of the trajectory? What is the specific model used here? "Enhanced VP" is too vague.

3.2. Please provide details about how the codes are generated.

4. Experimental evaluation should be improved.

4.1. In some tables, some of the models are not included without any explanations. Eg., it is not clear why UniVLA is not included in Table 2.

4.2. It is not clear why pi0.5 is not included in the evaluation at all.

4.3. A running time comparison with the VLA models should have been included.


Minor comments:
- "neural-symbolic paradigm" => paradigm of what? learning? programming? ..


**AFTER THE REBUTTAL**

The authors have addressed almost all of my concerns. The remaining bits: (1) The method is computationally more expensive compared to a VLA. (2) The approach introduces other challenges with the correctness of the generated code and the complexities of using external tools. (3) The neuro-symbolic claim, which has been admitted to be over-claimed by the authors, needs to be rephrased throughout the text.

I have increased my recommendation.

---

> ### Author Rebuttal · Authors · 2026-03-31
>
> **W1.1 (Computational overhead comparison):** We provide a computational-overhead comparison in our response to **Reviewer n1Yt’s W1**. Results show that **(1) NeurVLA achieves a favorable performance-overhead trade-off in ECoT construction,** with stronger performance at lower cost; **(2) NeurVLA remains efficient at deployment,** with competitive inference frequency and stronger performance.
>
> **W1.2 (Practicality):** As clarified in our response to **Reviewer n1Yt’s W1**, NeurVLA directly outputs actions at deployment, so its practical overhead is comparable to other VLA methods.
>
> **W1.3 (Infinite-loop prevention):** We set a maximum of 5 iterations in PTM. If this limit is reached, the current episode is terminated as a failure. As shown in our response to **Reviewer Zne9's Q2**, this limit is never reached in experiments.
>
> **W1.4 (Global correction):** "Fine-grained" refers to the precision of the correction rather than implying that only a very small local change can be made. If failure is more global, COG can still inject multiple precise corrective steps that substantially revise the plan.
>
> **W1.5 (Code reliability & verification):** We design different verification mechanisms for different errors:
> - **Syntax errors:** They are directly exposed during compilation/execution and can be corrected based on error logs.
> - **Semantic errors:** They can be detected automatically via property-based testing (e.g., PropTest [1]) and corrected according to the testing results.
> - Furthermore, semantic errors that are not detected typically lead to robot action failures, and the converted ECoTs are treated as negative samples for contrastive learning.
>
> |Code|Correct|Syntax Error|Syntax Correction Rate|Semantic Error|Semantic Correction Rate|
> |-|-|-|-|-|-|
> |COG|95.2%|1.3%|97.9%|3.5%|94.4%|
> |PTM|90.8%|2.0%|95.6%|7.2%|93.2%|
>
> **The results show that the generated code is highly reliable and can be effectively verified through the designed mechanisms.**
>
> ------
>
> **W2 (Clarification of neural-symbolic claim):**
> - We agree that "neural-symbolic" may be too strong in the classical symbolic-logic sense. Our intended emphasis is a programmatic reasoning framework combining neural modules with structured visual-program execution.
> - **For programmatic variables and operators,** the variables include objects, states, and trajectories, while the primitives include spatial-relation reasoning, trajectory prediction, and collision checking. The programs are grounded by explicit operators and geometric constraints rather than formal axioms.
> - **For terminology,** we appreciate the suggestion that "programmatic" may better characterize the framework and will revise the wording accordingly in the next version. More details on this discussion can be found in our response to **Reviewer n1Yt’s W2**.
>
> ------
>
> **W3.1 (Failure detection):** Based on the trajectory-augmented observation, NeurVLA first uses VP function `find()` (backed by **GLIP**) to localize task-relevant objects, and then applies functions `overlaps_with()` and `compute_depth()` (backed by **MiDaS**) to check overlap and estimate collision risk. As clarified in **Sec. 3.1** of the main text, "Enhanced VP" refers to standard VP augmented with functions for spatial relation reasoning and metric distance computation in the 3D coordinate system. We will include concrete code examples of these functions in the next version.
>
> **W3.2 (Code generation details):** During program generation, NeurVLA follows ViperGPT [2], feeding prompts and VP's function specifications to Code LLMs to automatically generate the programs. Concrete prompts are provided in **App. G**  of main text.
>
> ------
>
> **W4 (Experimental evaluation):**
> - **For Table 2,** due to page limits, we could not include every model on every benchmark in the main text. We provide UniVLA results on SimplerEnv-WidowX in our response to **Reviewer n1Yt’s W1**.
> - **For π0.5,** it was not open-sourced when we conducted the original experiments. We evaluate it on BEHAVIOR-100, RoboCasa, and additional real-world experiments on another robot platform, as detailed in our responses to **Reviewer Zne9’s W2** and **Reviewer dVNf’s W4**.
> - **For running-time comparison,** we compare NeurVLA with UniVLA in terms of inference time and task performance, as shown in our response to **Reviewer n1Yt’s W1**.
>
> ------
>
> **Minor comment:** NeurVLA is intended as a programmatic reasoning paradigm for failure-handling capability construction and internalization. In the next version, we will replace the original term with this more precise expression.
>
> **Limitations:** In the next version, we will add a paragraph on NeurVLA's key limitations and real-world deployment risks, including failure under distribution shift and the need for human oversight and safeguards.
>
> [1] Koo J, et al. PropTest: Automatic property testing for improved visual programming. 2024.
>
> [2] Suris D, et al. Vipergpt: Visual inference via python execution for reasoning. 2023.

---

> > ### Author Rebuttal · Reviewer_Tfgc · 2026-04-01
> >
> > The authors have addressed almost all of my concerns. The remaining bits: (1) The method is computationally more expensive compared to a VLA. (2) The approach introduces other challenges with the correctness of the generated code and the complexities of using external tools. (3) The neuro-symbolic claim, which has been admitted to be over-claimed by the authors, needs to be rephrased throughout the text.

---

> > > ### Author Response · Authors · 2026-04-02
> > >
> > > Thank you very much for reviewing our response and updating the assessment! We greatly appreciate your valuable comments and feedback! We are encouraged that our rebuttal has addressed most of your concerns, and we would like to further clarify the remaining three points you raised.
> > >
> > > **(1) Computational overhead:**
> > >
> > > **1.1 NeurVLA is a two-stage framework designed to enhance failure-handling capability of existing VLA models,** consisting of an introspective ECoT construction pipeline and a VLA training method. It first generates and executes programs for failure correction and prevention to construct introspective ECoTs, and then uses these ECoTs to train the VLA model to internalize such failure-handling capabilities. The trained VLA model directly outputs actions at deployment time. Therefore, the main computational cost of NeurVLA arises from the construction of introspective ECoTs.
> > >
> > > **1.2 For ECoT construction pipeline,** we compare NeurVLA with other ECoT construction methods in terms of the performance-overhead trade-off.
> > >
> > > |# Samples|Direct ECoT(MLLM) SR|Direct ECoT(MLLM) Time(h)|Code→ECoT(Seed-Coder-8B) SR|Code→ECoT(Seed-Coder-8B) Time(h)|Code→ECoT(GPT-5) SR|Code→ECoT(GPT-5) Time(h)|Direct ECoT(Human) SR|Direct ECoT(Human) Time(h)|
> > > |-|-|-|-|-|-|-|-|-|
> > > |2k|54.8|1.2|62.3|1.2|64.7|2.1|67.3|15.4|
> > > |4k|60.2|2.3|67.2|2.5|70.4|4.3|78.1|30.8|
> > > |6k|63.1|3.5|71.8|3.8|75.5|6.5|81.8|46.2|
> > > |8k|64.3|4.6|75.8|5.1|79.4|8.7|82.3|61.6|
> > > |10k|64.5|5.7|79.6|6.4|81.7|10.8|82.5|77.0|
> > >
> > > The results show that:
> > > - NeurVLA achieves a favorable performance-overhead trade-off during ECoT construction.
> > > - The overhead can be substantially reduced by using smaller open-source models with only limited performance degradation.
> > > - As data scales, the additional overhead is consistently translated into meaningful performance gains.
> > >
> > > **1.3 For deployment-time inference,** we further compare NeurVLA with UniVLA and π0.5 in both simulation and real-world settings in terms of inference frequency and task performance.
> > >
> > > ||SimplerEnv Inference Freq.(Hz)|Google Robot Avg. VM|Google Robot Avg. VA|WidowX Avg. Grasp SR|WidowX Avg. Task SR|Real-World Inference Freq.(Hz)|Real-World Task SR|
> > > |-|-|-|-|-|-|-|-|
> > > |UniVLA|12.1|39.2|49.4|71.5|48.0|10.2|32.8|
> > > |π0.5|15.0|80.3|81.1|90.3|65.9|12.5|70.5|
> > > |NeurVLA|11.8|88.7|90.0|94.8|70.4|9.9|77.9|
> > >
> > > The results show that:
> > > - NeurVLA maintains deployment-time inference frequency comparable to other VLA models.
> > > - By internalizing failure-handling capabilities from introspective ECoTs, NeurVLA achieves substantially stronger performance in both simulation and real-world settings.
> > >
> > > ------
> > >
> > > **(2) Code correctness and external-tool complexity:**
> > >
> > > **2.1 For generated code correctness,** we mitigate this challenge through designed verification and correction mechanisms:
> > > - **Syntax errors:** They are directly exposed during compilation/execution and corrected based on error logs. They account for only 1.7% of the generated code on average, with an average correction rate of 96.8%.
> > > - **Semantic errors:** They are detected automatically via property-based testing and corrected according to the testing results. They account for 5.4% of the generated code on average, with an average correction rate of 93.8%.
> > > - Furthermore, semantic errors that remain undetected typically lead to robot action failures, and the converted ECoTs are treated as negative samples for contrastive learning.
> > >
> > > **2.2 For the complexity introduced by external modules,** we would like to clarify two points:
> > > - **Manageable complexity:** External modules in NeurVLA, such as 3D-MOOD and ATM, are lightweight, practically manageable, and required only during introspective ECoT construction rather than deployment-time inference.
> > > - **Mitigated error propagation:** NeurVLA is highly robust to error propagation across external modules through three mechanisms:
> > >   **(i) Geometric consistency checks.** The program verifies the reasonableness of the 3D coordinates returned by 3D-MOOD before using them in subsequent reasoning.
> > >   **(ii) PTM-based prediction and verification.** PTM predicts future trajectories and checks whether they would lead to potential failures, allowing imperfect corrective steps caused by inaccurate 3D-MOOD outputs to be revised until the predicted execution is safe.
> > >   **(iii) Iterative recovery.** If imperfect corrective steps still lead to execution failure, NeurVLA both proceeds to the next iteration for post-execution correction and re-runs the current iteration with execution history and failure observations to revise the steps.
> > >
> > > As shown in our response to **Reviewer dVNf’s W1**, adding noise to external modules causes only a 1.9% relative performance drop, indicating strong robustness in practice.
> > >
> > > ------
> > >
> > > **(3) Rephrasing the neuro-symbolic claim:**
> > >
> > > We agree that the current wording over-claims this point. In the revision, we will rephrase it throughout the paper and describe NeurVLA more precisely as a programmatic reasoning framework.

---

### Official Review · Reviewer_Zne9 · 2026-03-08

**Soundness:** 3
**Presentation:** 3
**Significance:** 2
**Originality:** 2
**Overall Recommendation:** 4
**Confidence:** 4

**Summary:**

The paper proposes NeurVLA, a neural-symbolic reasoning framework that addresses 1) failure correction and 2) failure prevention for VLM models using 'Observation-grounded Failure Correction with Trajectory-Modeling Failure Prevention'.

NeurVLA also employs Reasoning-Guided Action Learning to internalize the failure-handling capabilities, which are from the neural-symbolic reasoning processes.

Experimental results show that the proposed model outperforms existing models in LIBERO, SimplerEnv and CALVIN.

**Compliance With Llm Reviewing Policy:**

Affirmed.

**Final Justification:**

My concerns have been adequately addressed. I decided to raise my score to weak accept.

**Key Questions For Authors:**

Q1. how accurate are 3D-MOOD and ATM? How much error propagation comes from these external modules?

Q2. how many iterations in average are until no collision is detected in Trajectory-Modeling Failure Prevention?

Q3. how many trajectories do you need to internalize failure-handling capability (size of data for SFT and contrastive learning)? did you find any trend the performance increase/decrease based on the data scale? I think this is important to apply the approach to real-world applications

**Limitations:**

Please see Weaknesses.

**Strengths And Weaknesses:**

Strengths:
1) the paper is well-written, easy to follow, presentations is good
2) the proposed methods are reasonable to tackle the addressed challenges
3) experiment results are strong.

Weaknesses:
1) benchmark limitation A: LIBERO, SimplerEnv and CALVIN, are static environments. In terms of what 'failure handling' is for, it should be tested on dynamic environments as well. The author also mentioned handling "unexpected environmental changes" is important, but not tested.
2) benchmark limitation B: benchmarks' performances are already saturated in many existing papers, also they are limited to the table-top tasks. would be interesting how the proposed method can perform on outside of table top tasks (e.g., BEHAVIOR-100, RoboCasa etc)
3) concerns about generalization: I wonder if the code generations (`check_previous_step.py` and `predict_plan_failure.py`) would generalize to more complex environments besides table-top tasks.
4) Overall, the method appears to work well within the benchmarks and 4 tasks with SO-ARM101, but I'm not yet convinced that is is a compelling approach for general robotics/VLA applications. I would be open to revising review if the authors can address these questions.

---

> ### Author Rebuttal · Authors · 2026-03-31
>
> **W1 (Dynamic environments/unexpected environmental changes):**
>
> We introduce four types of changes on a subset of RoboCasa, with navigation-dependent tasks excluded, to simulate dynamic environments and unexpected environmental changes. Each of these changes potentially causes robot execution failures.
> - Target-object state change: Goal object itself is moved or reoriented.
> - Task-relevant container state change: A task-relevant drawer, receptacle, or container state is changed.
> - Obstacle insertion: A new obstacle is introduced into the planned execution path.
> - Instruction change: Task instruction is modified.
>
> |Change Type|Backbone Task SR|CollabVLA Task SR|NeurVLA Task SR|Failure Correction Rate|Prevention Success Rate|
> |-|-|-|-|-|-|
> |Target-object state change|32.9|42.7|56.1|95.0|85.8|
> |Task-relevant container state change|21.6|30.0|54.7|84.0|80.1|
> |Obstacle insertion|28.3|37.8|55.3|80.0|89.4|
> |Instruction change|16.1|24.6|53.2|72.0|79.1|
>
> The results show that:
> - **NeurVLA performs well on dynamic environments.** Across the four change types, NeurVLA improves the average task success rate over the backbone by 30.1 and outperforms CollabVLA by 21.05.
> - **NeurVLA effectively handles unexpected environmental changes.** Besides achieving strong overall task success, it maintains at least 72.0 failure correction rate and 79.1 prevention success rate across the four change types, showing its ability to both recover from and prevent failures caused by unexpected environmental changes.
>
> ------
>
> **W2 (Beyond-tabletop benchmarks):**
>
> To assess whether NeurVLA generalizes beyond tabletop tasks, we evaluate it on subsets of BEHAVIOR-100 and RoboCasa after excluding navigation-dependent tasks, and compare it against UniVLA and π0.5. We further evaluate it on two **real-world beyond-tabletop tasks**, *Loading Dishwasher* and *Packing Suitcase*.
>
> |Method|BEHAVIOR-100 Q-Score|RoboCasa SR|Dishwasher SR|Suitcase SR|
> |-|-|-|-|-|
> |π0.5|0.55|52.9|27.8|30.2|
> |UniVLA|0.48|44.8|60.3|65.5|
> |NeurVLA|0.59|56.5|67.4|71.9|
>
> **The results show that NeurVLA performs well on tasks beyond tabletop settings,** it outperforms π0.5 and UniVLA by 16.6% and 53.6%, respectively.
>
> ------
>
> **W3 (Code generation generalization):**
>
>  `check_previous_step.py` is used for post-execution checking and injects corrective steps into subsequent plan when failure is detected, while `predict_plan_failure.py` is used for pre-execution failure prediction and adjusts the plan when a potential failure is predicted. As shown in **W1**, NeurVLA achieves strong failure correction and prevention rates, indicating that both modules remain effective in more complex environments beyond tabletop tasks.
>
> ------
>
> **W4 (General robotics/VLA applications):**
>
> Beyond the above experiments, we further conduct real-world experiments on another robot platform to evaluate the generality and applicability of NeurVLA, as detailed in our response to **Reviewer dVNf's W4**. The results show that:
>
> - NeurVLA handles representative daily-life tasks well.
> - NeurVLA generalizes across different robot platforms.
>
> ------
>
> **Q1 (3D-MOOD / ATM accuracy and error propagation):**
>
> **The standalone accuracy of 3D-MOOD and ATM is 96.2% and 92.5%, respectively.**
> For the analysis of error propagation from these external modules, please refer to our response to **Reviewer dVNf’s W1.**
>
> ------
>
> **Q2 (Average number of PTM iterations):**
>
> We summarize the percentage of cases requiring different numbers of refinement iterations in PTM until no collision is detected, as shown below. Here, "# Iterations = 1" means that PTM detects no collision for the initial subsequent plan, so no further refinement or detection is needed. The maximum number of PTM iterations across all cases is 4.
>
> |# Iterations|1|2|3|4|
> |-|-|-|-|-|
> |Percentage|46.2%|37.7%|15.2%|0.9%|
>
> **On average, PTM requires only 1.7 iterations, showing that the failure-prevention process is efficient in practice.**
>
> ------
>
> **Q3 (Data scale for internalizing failure-handling capability):**
>
> We synthesize **2k** demonstrations to internalize failure-handling capability, substantially smaller than the data used by other robust VLA methods such as CollabVLA (50k) and FailSafe (131k). To study data scaling, we train NeurVLA with different numbers of synthesized demonstrations, as shown below:
>
> |# Demonstrations|1k|2k|3k|4k|5k|6k|7k|8k|
> |-|-|-|-|-|-|-|-|-|
> |Avg. SR on SimplerEnv-WidowX|62.3|64.7|67.5|70.4|72.8|75.5|77.3|79.4|
>
> The results show that:
>
> - **NeurVLA improves steadily with more data.** As the number of synthesized demonstrations increases from 1k to 8k, its average success rate on SimplerEnv-WidowX consistently improves from 62.3 to 79.4.
> - **NeurVLA remains data-efficient at smaller scales.** Even with only 2k synthesized demonstrations, it already achieves performance comparable to robust VLA methods such as CollabVLA, which requires 50k training demonstrations.

---

> > ### Author Rebuttal · Reviewer_Zne9 · 2026-04-06
> >
> > All my concerns are well resolved. I decided to raise my score to weak accept.

---

> > > ### Author Response · Authors · 2026-04-06
> > >
> > > Thank you very much for reviewing our response and updating the assessment! We greatly appreciate your valuable comments and feedback!

---

### Official Review · Reviewer_dVNf · 2026-03-11

**Soundness:** 3
**Presentation:** 3
**Significance:** 2
**Originality:** 3
**Overall Recommendation:** 4
**Confidence:** 3

**Summary:**

This paper proposes NeurVLA, a neural-symbolic framework that enhances the failure-handling capability of Vision-Language-Action (VLA) models through two complementary mechanisms: Observation-Grounded Failure Correction (COG), which uses executable programs to precisely verify action completion and inject fine-grained corrective steps, and Trajectory-Modeling Failure Prevention (PTM), which explicitly models action-induced motion trajectories to detect and avoid potential failures before execution. These two mechanisms are integrated within a Reasoning-Guided Action Learning (LRG) pipeline that converts the neural-symbolic reasoning traces into Embodied Chain-of-Thought (ECoT) data for learning, thereby internalizing failure-handling as an intrinsic capability of the VLA model. Experiments across LIBERO, SimplerEnv, CALVIN, and real-world tasks demonstrate consistent improvements over baselines.

**Compliance With Llm Reviewing Policy:**

Affirmed.

**Final Justification:**

All my concerns are well resolved. I decided to maintain my score to weak accept.

**Key Questions For Authors:**

see the weaknesses

**Limitations:**

yes

**Strengths And Weaknesses:**

**Strengths**

1) The decomposition of failure handling into correction and prevention is well-motivated and clearly articulated. The two components address distinct and complementary aspects of the problem.

2) The use of executable neural-symbolic programs to ground failure analysis in precise, metric-level computation is a reasonable design choice that offers more interpretable and verifiable reasoning compared to purely end-to-end VLM-based reflection.

3) The experiment is extensive. The paper evaluates on three simulated benchmarks (LIBERO, SimplerEnv, CALVIN) and four real-world tasks, applies NeurVLA to two backbone VLA models with different action modeling paradigms.

**Weaknesses**

1) The framework relies on several external specialist models (3D-MOOD for monocular 3D detection, ATM for trajectory prediction, a VLM for ECoT conversion, and an LLM for code generation). While the paper discusses scalability in principle, the practical robustness and error propagation across these components are not thoroughly analyzed. If any individual module (e.g., 3D-MOOD) produces inaccurate outputs, the downstream reasoning could be systematically misled, yet the paper does not provide failure-mode analysis of the neural-symbolic pipeline itself.

2) The paper acknowledges that NeurVLA's performance degrades under image shifts and rotations, which distort coordinate-based checks. This is a notable limitation for real-world deployment where camera calibration may drift or viewpoints may change, and it somewhat undermines the claimed robustness advantage. A more detailed discussion of when the coordinate-grounded approach breaks down would strengthen the paper.

3) The Reasoning-Guided Action Learning component is described at a relatively high level, with key details deferred to the appendix (e.g., the contrastive learning formulation, the ECoT conversion prompts). Given that this is one of the three main contributions, the main paper should provide sufficient detail to assess its soundness independently. For instance, how the contrastive pairs are constructed and how the three training formats (Think, Act, Follow) are balanced are not clearly specified.

4) The real-world experiments, while showing large improvements, are conducted on a single robot platform with a head-mounted camera. The generalizability of the approach to different robot morphologies, sensor configurations, or more dexterous platforms remains unvalidated. The improvement over the backbone is striking but the absolute success rates of the baselines are quite low, which may partly reflect the baselines being poorly suited to the specific setup rather than a fundamental advantage of NeurVLA.

---

> ### Author Rebuttal · Authors · 2026-03-31
>
> **W1 (Error propagation):**
>
> **(1) Clarification on the Role of the External Modules.** We clarify that the neural-symbolic pipeline, including 3D-MOOD and ATM, is used to construct ECoTs for training rather than for online deployment. Moreover, we only convert programs that lead to successfully executed actions into ECoTs, so errors from these two modules do not accumulate to cause execution failures during deployment.
>
> **(2) Mechanisms for mitigating error propagation.** NeurVLA mitigates error accumulation in three ways:
> - **Geometric consistency checks.** When 3D-MOOD returns 3D coordinates, we use the program to perform geometric consistency checks on them. Only reasonable coordinates are used in subsequent reasoning.
> - **PTM-based prediction and verification.** PTM predicts future trajectories and verifies whether they would cause potential failures, so imperfect corrective steps caused by inaccurate 3D-MOOD outputs can still be revised until the predicted execution is safe.
> - **Iterative recovery.** If imperfect corrective steps still lead to execution failure, NeurVLA both proceeds to the next iteration for post-execution correction and re-runs the current iteration with execution history and failure observation to revise the steps.
>
> We further conducted an experiment to validate these mechanisms.
> |Setting|SR with 3D-MOOD/ATM|SR with Noisy 3D-MOOD/ATM|
> |-|-|-|
> |Original|91.2|89.5|
> |w/o Geom. Check|86.8|76.4|
> |w/o PTM Verif.|79.9|63.7|
> |w/o Iter. Recovery|84.6|71.2|
>
> These results show that error propagation does exist, but NeurVLA’s design mechanisms can substantially mitigate it and thereby improve robustness.
>
> **(3) Learning from Failed Iterations for More Robust Failure Handling.** Furthermore, we introduce LRG, where failed iterations are also converted into ECoT-action instances, providing contrastive supervision signals that help the model internalize more robust failure-handling ability.
>
> ------
>
> **W2 (Breakdown analysis):**
>
> **(1) Analysis of when NeurVLA breaks down under image shifts and rotations.**
> We analyze when NeurVLA breaks down under different image shift magnitudes and rotation angles, and compare it with CollabVLA, a robust VLA baseline.
>
> Image shift (SR):
> | |0|±0.10H/W|±0.20H/W|±0.30H/W|
> |-|-|-|-|-|
> |CollabVLA|90.8|69.1|8.3|0.0|
> |NeurVLA|96.7|88.4|52.5|0.0|
>
> Image rotation (SR):
> | |0°|±10°|±20°|±30°|
> |-|-|-|-|-|
> |CollabVLA|91.0|80.3|49.7| 0.0  |
> |NeurVLA|96.4|90.2|64.6|19.3|
>
> **(2) Analysis of when NeurVLA breaks down under diverse perturbations.**
> We further provide broader robustness comparisons against both the backbone and CollabVLA in **App. E** across **action, observation, environment, and instruction** perturbations, where image shifts and rotations are included under observation perturbations. Overall, the results show that:
>
> - **For breakdown thresholds,** under image shifts and rotations, NeurVLA breaks down at ±0.30H/W and ±30°, respectively; under broader perturbations, it breaks down at 15% noise level for action perturbations and 43% noise level for observation perturbations.
> - **For robustness,** averaged across the four perturbation types, NeurVLA remains more robust than both the backbone and CollabVLA, tolerating larger perturbations before breakdown.
>
> ------
>
> **W3 (Details of LRG):**
>
> Given the observation $o$ and instruction $inst$, NeurVLA decomposes task completion into multiple iterations, each consisting of an ECoT $e$ and an action $a$. For each iteration, we instantiate three training formats: Think $(o,inst)→(e,a)$, Act $(o,inst)→a$, and Follow $(o,inst,e)→a$. Since all formats are constructed from each iteration, they are naturally balanced in the training data. For an execution-failed iteration $(o,inst,e^{-},a^{-})$, NeurVLA re-runs it by feeding back the execution history and failure observation, where PTM revises the plan to avoid repeating the same failure, yielding a successful iteration $(o,inst,e^{+},a^{+})$. These two iterations form a contrastive pair. We thank the reviewer for this suggestion and will move these key training details to the main text in the next version.
>
> ------
>
> **W4 (Real-world generalizability):**
>
> We further conduct real-world experiments on a Unitree G1 robot with a Dex-3 manipulator, using the same tasks as in the main text and comparing with a stronger baseline, π0.5.
> | |Blocks Drawer|Towel Fold|Pour Ball|3-Disc Hanoi|
> |-|-|-|-|-|
> |π0.5|81.1|82.4|84.0|34.6|
> |NeurVLA|86.9|88.7|86.5|49.3|
>
> The results show that:
> - **NeurVLA handles representative daily-life tasks well.** These tasks cover object pick-and-place, deformable-object manipulation, precise control, and long-horizon reasoning, and their difficulty mainly arises from the intrinsic need for reasoning and failure handling.
> - **NeurVLA generalizes across different robot platforms.** It achieves an average success rate about 10.4% higher than the strongest baseline, π0.5, further supporting its deployability rather than reflecting setup-specific bias.

---

> > ### Author Rebuttal · Reviewer_dVNf · 2026-04-01
> >
> > All my concerns are well resolved. I decided to maintain my score to weak accept.

---

> > > ### Author Response · Authors · 2026-04-01
> > >
> > > Thank you very much for reviewing our response. We greatly appreciate your valuable comments and feedback.

---

### Decision · Program_Chairs · 2026-04-30

**Decision:**

Accept (regular)

**Comment:**

This paper proposes a training-free-at-deployment framework for handling VLA failures by decomposing the problem into (a) post-execution correction and (b) pre-execution prevention both grounded in executable visual programs with geometric primitives. These mechanisms are distilled into the base VLA via embodied CoT training, enabling the deployed model to act directly without external modules. The framing is clean and practical, and the use of programmatic reasoning to unify correction and prevention is well-motivated.

All reviewers rated the paper as weak accept (4/4/4/4, confidences 3–4), consistently highlighting the decomposition and strong empirical coverage (LIBERO, SimplerEnv, CALVIN, real-world tasks, two VLAs). Main concerns included potential error propagation from the multi-module pipeline, computational overhead, code-generation reliability, and overclaiming “neural-symbolic” reasoning. The rebuttal substantially strengthened the paper with new experiments (RoboCasa, BEHAVIOR-100, real-world tasks beyond tabletop, Unitree G1), stronger baselines (pi_0.5), and targeted analyses (noise injection, code correctness, PTM iterations). The authors also agreed to reframe “neural-symbolic” as “programmatic reasoning,” resolving a key terminology concern.

While the contribution is primarily systems-oriented, it is well-executed: the correction/prevention decomposition is principled, and distillation of the offline pipeline allows efficient deployment of the model. The evaluation is thorough and the rebuttal’s analyses added additional value.